# Clinical Course and Risk Factors for In-Hospital Mortality of 205 Patients with SARS-CoV-2 Pneumonia in Como, Lombardy Region, Italy

**DOI:** 10.3390/vaccines9060640

**Published:** 2021-06-11

**Authors:** Mauro Turrini, Angelo Gardellini, Livia Beretta, Lucia Buzzi, Stefano Ferrario, Sabrina Vasile, Raffaella Clerici, Andrea Colzani, Luigi Liparulo, Giovanni Scognamiglio, Gianni Imperiali, Giovanni Corrado, Antonello Strada, Marco Galletti, Nunzio Castiglione, Claudio Zanon

**Affiliations:** 1Department of Medicine, Division of Hematology, Valduce Hospital, 22100 Como, Italy; agardellini@valduce.it; 2Risk Management, Valduce Hospital, 22100 Como, Italy; lberetta@valduce.it; 3Quality Management System, Valduce Hospital, 22100 Como, Italy; lbuzzi@valduce.it; 4Department of Medicine, Division of Pneumology, Valduce Hospital, 22100 Como, Italy; sferrario@valduce.it (S.F.); acolzani@valduce.it (A.C.); 5Department of Mother and Child, Division of Paediatric and Neonatology, Infectious Diseases Consultant, Valduce Hospital, 22100 Como, Italy; svasile@valduce.it; 6Department of Medicine, Division of Neurology, Valduce Hospital, 22100 Como, Italy; rclerici@valduce.it; 7Department of Medicine, Division of Internal Medicine, Valduce Hospital, 22100 Como, Italy; lliparulo@valduce.it; 8Department of Medicine, Division of Oncology, Valduce Hospital, 22100 Como, Italy; giosco@valduce.it; 9Department of Medicine, Division of Gastroenterology, Valduce Hospital, 22100 Como, Italy; g.imperiali@libero.it; 10Department of Medicine, Division of Cardiology, Valduce Hospital, 22100 Como, Italy; gcorrado@valduce.it; 11Department of Emergency Medicine, Valduce Hospital, 22100 Como, Italy; astrada@valduce.it; 12Department of Intensive Care Medicine, Valduce Hospital, 22100 Como, Italy; mgalletti@valduce.it; 13Hospital Health Management, Valduce Hospital, 22100 Como, Italy; ncastiglione@valduce.it (N.C.); zanonclaudio762@gmail.com (C.Z.)

**Keywords:** SARS-CoV 2, coronavirus disease 2019, pneumonia, therapy, treatment, mortality, risk factors

## Abstract

The aim of this study is to explore risk factors for in-hospital mortality and describe the effectiveness of different treatment strategies of 205 laboratory-confirmed cases infected with SARS-CoV-2 during the Lombardy outbreak. All patients received the best supportive care and specific interventions that included the main drugs being tested for repurposing to treat COVID-19, such as hydroxychloroquine, anticoagulation and antiviral drugs, steroids, and interleukin-6 pathway inhibitors. Clinical, laboratory, and treatment characteristics were analyzed with univariate and multivariate logistic regression methods to explore their impact on in-hospital mortality. Univariate analyses showed prognostic significance for age greater than 70 years, the presence of two or more relevant comorbidities, a P/F ratio less than 200 at presentation, elevated LDH (lactate dehydrogenase) and CRP (C-reactive protein) values, intermediate- or therapeutic-dose anticoagulation, hydroxychloroquine, early antiviral therapy with lopinavir/ritonavir, short courses of steroids, and tocilizumab therapy. Multivariable regression confirmed increasing odds of in-hospital death associated with age older than 70 years (OR 3.26) and a reduction in mortality for patients treated with anticoagulant (−0.37), antiviral lopinavir/ritonavir (−1.22), or steroid (−0.59) therapy. In contrast, hydroxychloroquine and tocilizumab have not been confirmed to have a significant effect in the treatment of SARS-CoV-2 pneumonia. Results from this real-life single-center experience are in agreement and confirm actual literature data on SARS-CoV-2 pneumonia in terms of both clinical risk factors for in-hospital mortality and the effectiveness of the different therapies proposed for the management of COVID19 disease.

## 1. Introduction

Severe acute respiratory syndrome coronavirus 2 (SARS-CoV-2), the cause of coronavirus disease 2019 (COVID-19), represents a viral disease infecting millions of individuals all over the world and has emerged as a major public health emergency of international concern. Therefore, on 11 March 2020, the World Health Organization (WHO) declared COVID-19 a pandemic disease because of widespread infectivity and high contagion rates. Full-genome sequencing indicated that COVID-19 is a betacoronavirus in the same subgenus as the severe acute respiratory syndrome (SARS) virus. The structure of the receptor-binding gene region is very similar to that of the SARS coronavirus, and the virus uses the same receptor, the angiotensin-converting enzyme 2 (ACE2), for cell entry [1]. Direct person-to-person transmission is the primary means of transmission of SARS-CoV-2, mainly through close-range contact via respiratory droplets or by transfer to mucous membranes after coming into contact with contaminated surfaces; 2 and 12 days are the lower and upper extremes of the incubation period for COVID-19, with most cases occurring approximately 4 to 5 days after exposure [2,3]. Several studies describing the clinical features of COVID-19 have been performed on hospitalized populations [4,5,6]. The most common clinical characteristics at the onset of the disease were fever (even low-grade fever <38 °C), cough, fatigue, and dyspnea with typical bilateral infiltrates on chest imaging. Other features, such as upper respiratory tract symptoms, myalgias, diarrhea, abdominal pain, anosmia, and dysgeusia, may also be present [7,8,9,10]. The spectrum of symptomatic infection ranges from mild to critical, with a proportion of severe or critical disease reported in approximately 20% of cases [5,11,12]. Acute respiratory distress syndrome (ARDS) is the major complication in patients with severe disease and can manifest shortly after the onset of dyspnea [4,12]. Other common complications include thromboembolic disorders, cardiovascular disease (e.g., arrhythmias, acute cardiac injury, and shock), an exuberant inflammatory response similar to cytokine release syndrome, and secondary infections [4,13,14,15,16,17,18]. Severe illness can occur in otherwise healthy individuals of any age, but the highest proportion of severe cases occurs in adults older than 60 years of age or presenting underlying medical comorbidities. Potential risk factors for severe illness include cardiovascular and cerebrovascular diseases, diabetes mellitus, chronic lung disease, malignancies, and obesity [19,20,21,22]. In addition, a significantly higher prevalence of COVID-19 disease and increased mortality have been recorded in patients with an impairment of the immune response, such as onco-hematological patients or individuals with autoimmune diseases [23,24]. The most common laboratory abnormalities among hospitalized patients with COVID-19 include lymphopenia, elevated aminotransaminase, lactate dehydrogenase (LDH), and inflammatory markers such as C-reactive protein (CRP), ferritin, and the erythrocyte sedimentation rate (ESR) [3,4,6,25,26,27]. Common abnormal chest radiograph findings were consolidation and ground-glass opacities, with bilateral, peripheral, and lower lung zone distributions [28]. Chest computed tomography (CT) studies confirmed the presence of bilateral peripheral ground-glass opacification, with or without consolidative abnormalities, while less common findings included pleural thickening, pleural effusion, and lymphadenopathy [29,30,31].

A number of approaches to COVID-19 therapy have been investigated, including chloroquine, antiviral drugs, interleukin-6 pathway inhibitors, and other immunomodulators. In vitro studies have shown that hydroxychloroquine and chloroquine can bind cell surface sialic acid and gangliosides with high affinity, thereby impairing SARS-CoV-2 from binding to host cell angiotensin-converting enzyme (ACE)-2 receptors [32,33]. On the basis of their potential antiviral activity in vitro, several randomized trials have been conducted to evaluate their clinical use, but none have suggested a clear efficacy [34,35,36]. In addition, most observational studies have not suggested a benefit with chloroquine or hydroxychloroquine treatment [37,38] and have highlighted the potential risk for toxicity of those drugs. Arrhythmias and QT interval prolongation emerged as the most relevant adverse events, particularly when these agents are administered in patients with comorbidities or in combination with other medicines known to prolong the QT interval, including azithromycin. Antiviral agent lopinavir/ritonavir has been considered a promising treatment option for COVID-19 infections based on its proven in vitro efficacy against other novel coronaviruses SARS-COV via the inhibition of 3-chymotrypsin-like protease [39,40]. However, results from a randomized trial did not demonstrate a clear benefit of lopinavir–ritonavir compared with standard care alone at the cost of increased toxicity [41]. Due to the observation that some patients have a marked elevation in proinflammatory markers, with a clinical presentation that resembles cytokine release syndrome, interrupting the inflammatory cascade has been proposed as a potential therapeutic target for COVID-19 to prevent disease progression. Tocilizumab, an interleukin (IL)-6 receptor inhibitor, is being evaluated in randomized trials for the treatment of COVID-19 [42,43,44]. Similarly, glucocorticoids have been used in patients with moderate to severe ARDS [45].

Here, we report the epidemiological, clinical, and laboratory characteristics and treatment and clinical outcomes of 205 laboratory-confirmed cases infected with SARS-CoV-2 admitted to Valduce Hospital in Como, Italy. The aim of this study is to explore risk factors for in-hospital mortality and describe the effectiveness of different treatment strategies in a retrospective single-center cohort study. Finally, we explore the potential predictive role of several surrogate markers of inflammation as predictors of negative outcomes for hospitalized patients, with the aim of developing an easy-to-use prognostic score to identify patients with poor prognoses at an early stage.

## 2. Materials and Methods

Briefly, 205 patients aged between 17 and 100 years (male/female: 113/92) were admitted to Valduce Hospital between March and April 2020 because of SARS-Cov-2 pneumonia, requiring hospitalization, and were included in this analysis; 168 patients came to the emergency room for direct observation, while 37 patients were sent to the hospital from a nursing home for the elderly. All patients had a laboratory-confirmed SARS-CoV-2 infection through reverse transcriptase-polymerase chain reaction (RT-PCR) on nasopharyngeal swab specimens. For each patient, data regarding history, vital signs, ratio of arterial oxygen partial pressure to fractional inspired oxygen (P/F), need for oxygen therapy, blood chemistry parameters, treatment schedule, and outcome were recorded. Blood chemistry variables were recorded as both value at presentation and maximum value achieved during hospitalization, while interleukin 6 (IL6) values were recorded only in 29 patient candidates for tocilizumab therapy. Patients’ characteristics at presentation are shown in Table 1.

The median age at presentation was 77 years (interquartile range, IQR: 65–83), with 65 patients (31.7%) aged less than 70 years, 50 patients (24.4%) aged 70 to 79 years, 70 patients (34.1%) aged 80 to 89 years, and 20 patients (9.8%) aged 90 years or older. In addition, 35 patients (17%) presented no comorbidity, 52 patients (25.4%) had 1 comorbidity, and 118 patients (57.6%) had two or more comorbidities. Among the main comorbidities, there were arterial hypertension (*n =* 115), atrial fibrillation (*n* = 43), diabetes (*n =* 32), chronic obstructive pulmonary disease (*n =* 25), and obesity (*n* = 18).

All patients received the best supportive care and, based on their clinical needs and comorbidities, specific interventions that included the main drugs being tested for repurposing to treat COVID-19, such as hydroxychloroquine, anticoagulation and antiviral drugs, steroids, or interleukin-6 pathway inhibitors. Overall, hydroxychloroquine (HCQ) was administered to 160 patients (78%) with a loading dose of 400 mg twice daily, followed by maintenance with 200 mg twice daily for 7–14 days. Antiviral therapy was administered to 52 patients (25.3%) and consisted of lopinavir/ritonavir 400/100 mg bid for 5–7 days. Since only 2 patients were treated with remdesivir on a compassionate-use basis, this treatment was not included in the statistical analysis; 34 patients (16.5%) continued their usual anticoagulant therapy for previous medical conditions, while 124 patients (60.5%) were treated with low molecular weight heparin (LMWH) at intermediate doses of 100 U/kg/day. In addition, 29 patients were evaluated for anti-IL6 therapy based on their clinical characteristics and drug availability, and 21 patients (10.2%) received tocilizumab 8 mg/kg (up to a maximum of 800 mg per dose), with a second administration after 12 h on compassionate use; 90 patients (43.9%) with the need for oxygen therapy after more than 7 days from the onset of symptoms were treated with steroid therapy, consisting of methylprednisolone 1 mg/kg for 5 days and then tapered according to clinical evolution. Almost all patients (95.6%) were treated with a concomitant short course of antibiotic therapy, usually including azithromycin 500 mg per day for 3 to 6 days.

The need for oxygen therapy was distinguished based on the amount of oxygen delivered to maintain an adequate P/F ratio. Oxygen delivery systems are classified as low-flow or variable-performance devices and high-flow or fixed-performance devices. Low-flow systems provide oxygen at flow rates that are lower than patients’ inspiratory demands, where high-flow systems provide a constant FiO2 by delivering the gas at flow rates that exceed the patient’s peak inspiratory flow. Oxygen was delivered using nasal cannulas (*n* = 32), bag-valve masks (*n* = 36), Venturi masks (*n* = 41), or continuous positive airway pressure (CPAP, *n* = 64); 16 patients (7.8%) did not need oxygen therapy, while another 16 patients (7.8%) were transferred to the intensive care unit for invasive mechanical ventilation.

All collected variables were submitted to descriptive methods. The Mann–Whitney U-test was used for comparison of continuous non-normally distributed variables. Survival analysis was carried out using the Kaplan–Meier product-limit method, followed by the log-rank test, to evaluate the possible differences in survival between groups. Cox univariate and multivariate regression models were also used to analyze the effects of continuous variables on survivorship and evaluate the role of different clinical variables as predictors for in-hospital mortality. The following potential prognostic parameters were evaluated: age, sex, comorbidities, P/F ratio, oxygen therapy, white blood cell count (WBC), lactate dehydrogenase (LDH), C-reactive protein (CRP), procalcitonin (PCT), serum ferritin, D-dimer, interleukin 6 (IL6), alanine aminotransferase (ALT), and aspartate aminotransferase (AST). Blood chemistry variables were assessed for survival, both value at presentation and maximum value achieved during hospitalization. The optimal multivariate model was chosen using backward stepwise elimination after inserting all variables showing *p* < 0.05 at univariate analysis. The receiver operating characteristics curve (ROC) was traced to analyze the role of continuous variables on survivorship and to search for an optimal cut-off value for the variables themself. For all possible cut-off points, total accuracy was considered together with sensitivity, specificity, positive predictive value, and negative predictive value; however, the choice was made according to Youden.

Statistical significance was assumed for all tests with *p* < 0.05. Statistical analysis was done using MedCal statistical software version 9.3.7.0 (MedCalc Software, Ostend, Belgium).

## 3. Results

Two hundred and five patients were included in the analysis. The median follow-up time was 16 days based on the reverse Kaplan–Meier method. The median hospitalization time was 7 days (range: 1–27) and 12 days (range: 1–43) in deceased and surviving patients, respectively. Among the 107 surviving patients (52.2%), 58 were discharged from the hospital and 49 were transferred to rehabilitation facilities. Estimated 7-day and 28-day survival rates were 69.4% and 34.1%, respectively (Figure 1). Statistical analysis showed no significant difference in terms of clinical variables between male and female patients.

### 3.1. Prognostic Factors for Survival

Age distribution and P/F ratio at presentation showed an optimal cut-off point at 69 years (area under the curve, AUC 0.71, sensitivity 86.7%, specificity 48.6%, LR + 1.69, LR − 0.27; *p* = 0.0001) and 233 (AUC 0.73, sensitivity 65.4%, specificity 75.0%, LR + 2.62, LR − 0.46; *p* = 0.0001), respectively. Among the surrogate markers of inflammation, LDH and CRP showed an optimal cut-off point at 395 U/L (AUC 0.77, sensitivity 75.0%, specificity 71.2%, LR + 2.61, LR − 0.35; *p* = 0.0001) and 124 mg/L (AUC 0.70, sensitivity 80.3%, specificity 60.9%, LR + 2.05, LR − 0.32; *p* = 0.0001) respectively, while no significant cut-off points for WBC, PCT, ferritin, D-dimer, ALT, or AST were identified. For the 29 patients tested, IL6 values showed an optimal cut-off point at 3484 pg/mL (AUC 0.862, sensitivity 66.7%, specificity 95.6%, LR + 15.33, LR − 0.35; *p* = 0.0003).

At univariate analyses, age with a cut-off point set at 70 years showed prognostic significance for survival, with an estimated 28-day survival rate of 67.4% and 21.4% for patients aged less or greater than 70 years, respectively (log-rank test: *p* < 0.0001) (Figure 2A).

Univariate analyses for clinical variables showed prognostic significance for the number of relevant comorbidities (28-day survival rate: 61.8%, 51.7%, and 35.3% for none, 1, or 2 or more comorbidities; *p* = 0.0008), P/F ratio less than 200 at presentation (21-day survival rate: 14.7% vs. 52.4%; *p* < 0.0001), high levels of LDH (28-day survival rate: 26.4% vs. 65.3%; *p* = 0.0001), and elevated C-reactive protein (CRP) values (25.4% vs. 74.9%; *p* = 0.0001), while no statistical significance was found for all the other clinical variables tested (see Figure 3A–D).

Finally, univariate analysis was tested for survival for the different treatments scheduled (Figure 4A–F). Patients requiring only low-flow oxygen therapy experienced improved survival compared to patients deserving of high-flow therapy, with an estimated 28-day survival rate of 50.0% and 23.3%, respectively (*p* < 0.0001). The use of any anticoagulant treatment resulted in an improvement in survival (estimated 28-day survival rate: 37.5% vs. 23.8%; *p* = 0.0001), while no difference was recorded between intermediate-dose or therapeutic-dose anticoagulation (36.9% and 37.1%, respectively). Hydroxychloroquine treatment was administered to 160 patients, resulting in an estimated 28-day survival rate of 35.7% vs. 27.3% of those who did not receive such therapy (*p =* 0.0029). Fifty-two patients received early antiviral therapy with lopinavir/ritonavir, which resulted in an estimated 28-day survival rate of 60.1% vs. 22.4% (*p* < 0.0001); 90 patients were treated with steroids, and 21 patients received tocilizumab. Both anti-inflammatory treatments showed prognostic significance for survival, with an estimated 28-day survival rate between treated and nontreated patients of 47.9% vs. 18.2% (*p* < 0.0001) and 69.4% vs. 29.6% (*p* = 0.0059), respectively.

When combined in multivariate analysis with backward elimination of factors, multivariable regression confirmed increasing odds of in-hospital death associated with age older than 70 years (OR 3.26, 95% CI 1.81–5.86; *p* < 0.0001) and showed reduced odds of mortality in patients treated with intermediate- or therapeutic-dose anticoagulation (OR −0.37, 95% CI 0.49–0.95; *p* = 0.0273), antiviral drug lopinavir/ritonavir (OR −1.22, 95% CI 0.16–0.54; *p* < 0.0001), or steroids therapy (OR −0.59, 95% CI 0.35–0.87; *p* = 0.0117) (Cox proportional-hazards regression model: *p* < 0.0001) (Table 2).

### 3.2. Prognostic Score Model

Based on ROC curve analysis, inflammation markers CRP and LDH were incorporated into a prognostic score system, attributing 1 point for each value above the identified cut-off (nominally, 124 mg/L for CRP and 395 U/L for LDH). Having none, one, or both of the positive markers determined a reduction in the estimated 28-day survival rate from 94.0% to 51.3% and 21.0%, respectively (log-rank test: *p* < 0.0001) (Figure 5).

## 4. Discussion

At the end of 2019, a novel coronavirus was identified as the cause of a cluster of pneumonia cases in the region of Wuhan, China. The infection rapidly spread, resulting in an epidemic throughout China, followed by an increasing number of cases in other countries throughout the world. Individuals of any age can acquire SARS-CoV-2 infection, although middle-aged and older adults are most commonly affected [4,5]. In a modeling study based on Chinese data, the hospitalization rate for COVID-19 increased with age, with an 11.8% rate for those 60 to 69 years old, 16.6% rate for those 70 to 79 years old, and 18% for those older than 80 years [46]. Older age has also been associated with more severe disease and increased fatality rate, though there is no clear age cut-off point [11,22,47]. In a report from the Chinese Center for Disease Control and Prevention, mortality rates were 8% and 15% among those aged 70 to 79 years and 80 years or older, respectively, in contrast to the 2.3% case fatality rate for the entire cohort [11]. Similar findings were reported from Onder et al. in an Italian cohort, with case fatality rates of 12.8% and 20.2% for patients aged 70 to 79 years and 80 years or older, respectively [22]. 

Among the approximately 4000 confirmed infections reported to Centers for Disease Control and Prevention in the United States, mortality was confirmed highest among older individuals, with 80% of deaths occurring in persons over the age of 65 and the highest percentage of severe outcomes among those older than 85 years [48]. Our cohort of patients is consistent with this data, with a median age at presentation of 77 years and with 68.2% of patients aged 70 or older. Consistent with CDC data, mortality was confirmed highest among elderly patients, with 86.7% and 62.2% of deaths occurring in persons older than 70 and 80 years, respectively. Analysis of the ROC curve confirmed an optimal cut-off point at 69 years, with an estimated 28-day survival rate of 67.4% and 21.4% for patients younger or older than 70 years, respectively; age over 70 years proved to be an independent risk factor for survival when combined in multivariate analysis. Several comorbidities have been associated with severe illness and mortality in SARS-CoV-2 infection. In a subset of 355 Italian patients who died with COVID-19, the mean number of preexisting comorbidities was 2.7, with less than 1% of patients presenting no underlying condition and approximately half of the patients presenting 3 or more comorbidities [22]. In accordance with these data, in our study, 57.6% of patients had two or more preexisting comorbidities among those considered potential risk factors for severe COVID-19 disease. Survival analysis showed prognostic significance for comorbidities, but the significance was not retained when tested against other clinical and therapeutic variables in multivariate analysis.

Several laboratory features may be altered among hospitalized patients with COVID-19. High D-dimer levels and severe lymphopenia have been associated with critical illness, while procalcitonin levels on admission are more likely to be elevated in patients requiring ICU care [4,5,7]. Elevated lactate dehydrogenase, C-reactive protein, and interleukin-6 levels are other common laboratory findings among hospitalized patients with COVID-19 [3,4,6,26]. However, none of these characteristics have been clearly demonstrated to have a strong prognostic value. In our patient population, LDH, CRP, and IL-6 showed an optimal cut-off point for survival in a ROC curve analysis, while we found no significance for the other laboratory tests of WBC, PCT, ferritin, and D-dimer. Based on these data, inflammation markers CRP and LDH have been incorporated into a prognostic score system, which has been shown to recognize three distinct groups of patients characterized by different disease severities and clinical outcomes. Validation of this potential easy-to-apply clinical score is required on a larger prospective patient population.

The rapid diffusivity and high mortality of severe cases have led researchers and clinicians to experience the impact of new and old drugs for the treatment of SARS-CoV-2 infection. The optimal approach to the treatment of COVID-19 disease is uncertain since for most potential therapies, evidence of their use comes primarily from observational data or indirect evidence. Hydroxychloroquine (HCQ), notably used in autoimmune diseases, has been considered of potential interest in SARS-CoV-2 infection. In vitro studies have indeed shown that HCQ binds cell surface sialic acid and gangliosides with high affinity, thereby impairing SARS-CoV-2 spike protein recognition and binding to host cell ACE-2 receptors [32]. On the basis of these data, the Taiwanese CDC declared HCQ a potential important anti-SARS-CoV-2 agent on 26 March 2020. An early French study highlighted the ability of 600 mg daily of HCQ, particularly in combination with azithromycin, to clear respiratory viral loads in 3 to 6 days [34]. In a subsequent retrospective multicenter cohort study among 1438 hospitalized patients with a diagnosis of COVID-19, hospitalized in metropolitan New York, treatment with hydroxychloroquine, azithromycin, or both, compared with neither treatment, was not significantly associated with differences in in-hospital mortality [35]. Finally, in an open-label randomized trial of 150 hospitalized patients with mild to moderate COVID-19, adding hydroxychloroquine to the standard of care did not improve the rate of SARS-CoV-2 clearance nor did it result in symptomatic improvement by 28 days [36].

In accordance with these literature data, in our patient cohort, HCQ was safely administered to 160 patients, and no treatment-related adverse events were recorded. Although HCQ treatment showed prognostic significance for survival at univariate analysis, significance was not confirmed in the multivariate analysis (*p =* 0.146). Finally, several drug agencies from different countries now do not recommend the use of hydroxychloroquine or chloroquine outside the setting of a clinical trial, given the lack of clear benefit from limited data and potential for toxicity. A multicenter, adaptive, randomized, and open clinical trial of the safety and efficacy of treatments for COVID-19, including HCQ, in hospitalized adults is now ongoing (NCT04315948).

The combined protease inhibitor lopinavir/ritonavir, primarily used for HIV infection, has demonstrated an in vitro activity against SARS-CoV via the inhibition of 3-chymotrypsin-like protease [39,40]. In a multicentric retrospective matched-cohort study conducted during the SARS outbreak, the addition of lopinavir/ritonavir as initial treatment was associated with a statistically significant reduction in the overall death rate and intubation rate compared with matched controls [49]. In addition, a significant reduction in adverse events, such as severe respiratory deterioration and lower nosocomial infections, was also noted in patients treated with lopinavir/ritonavir, and significance was confirmed at multivariate analysis [39,49]. Notably, no published in vitro activity data against SARS-CoV-2 exist for lopinavir/ritonavir. Results from a randomized open-label trial involving hospitalized adult patients with confirmed SARS-CoV-2 infection did not demonstrate a clear benefit of lopinavir/ritonavir treatment in 199 patients with severe COVID-19 beyond standard care alone [41]. Though there was a trend towards decreased mortality (particularly when administered within 12 days of symptom onset) and a reduction in serious adverse events, treatment with lopinavir/ritonavir was not associated with a difference in the time to clinical improvement nor in mortality at 28 days. Furthermore, the timing of lopinavir/ritonavir administration during the early peak of the viral replication phase (initial 7–10 days) appears to be crucial since delayed therapy initiation showed no effect on clinical outcomes [49,50].

In our study, short-term antiviral therapy was administered to 52 patients early, on admission, and within 7 days of symptom onset. Furthermore, particular attention had been paid to possible drug–drug interactions, with only a few low-grade transient gastrointestinal events and no serious treatment-related adverse outcomes. At univariate analysis, antiviral therapy showed prognostic significance for survival, with a clear benefit in terms of 28-day survival. When combined in multivariate analysis, antiviral therapy proved to be an independent risk factor for survival. The WHO has recently launched the multinational SOLIDARITY trial to further evaluate the activity of these antiviral treatments in COVID-19 disease.

Markedly elevated inflammatory markers (e.g., D-dimer, ferritin) and elevated proinflammatory cytokines, such as interleukin (IL)-6, are associated with severe COVID-19 disease, and blocking the inflammatory pathway has been hypothesized to prevent disease progression [15]. Glucocorticoids were the first drugs introduced with this aim [5,7]. Based on data reporting the potential benefit of glucocorticoids in patients with moderate to severe ARDS, a weak recommendation in favor of such therapy has been provided in this population. However, glucocorticoid administration in critically ill patients with COVID-19-related ARDS is not routinely suggested since data are limited to a single retrospective Chinese cohort [12]; there is concern about delayed viral clearance in patients with ARDS due to viral pneumonia. In our cohort, glucocorticoids were administered in patients still needing oxygen therapy at the end of the viral replication phase (i.e., after more than 7 days from the onset of symptoms), which consisted of a short course of methylprednisolone, up to 1 mg/kg for 5 days with a rapid tapering. Patients who presented with symptoms or signs of bacterial overinfection were excluded from glucocorticoid treatment. At univariate analysis, corticoid therapy showed a clear survival benefit, with a rapid improvement of respiratory pattern; when combined in multivariate analysis, glucocorticoid treatment proved to be an independent risk factor for survival.

Tocilizumab is an IL-6 receptor inhibitor used for rheumatic diseases and cytokine release syndrome (CRS) in CAR-T therapy. Several case reports and observational studies have described the use of tocilizumab in patients with COVID-19 [51,52,53,54], and its benefits have been evaluated in a prospective open, single-arm multicenter study on patients with severe disease [55]. No major adverse events have been directly related to tocilizumab treatment, and its use has been associated with a decrease in inflammatory markers and an improvement in respiratory parameters overall, with a reduced number of ICU admissions. A prospective series of 100 consecutive COVID-19 Italian patients with pneumonia and ARDS confirmed that the response to tocilizumab was rapid, sustained, and associated with significant clinical improvement [56].

In our patient cohort, tocilizumab was administered for compassionate use to 21 patients with moderate to severe COVID-19 pneumonia. No evident treatment-related adverse events were recorded, and although tocilizumab infusion showed a prognostic significance for 28-day survival at univariate analysis, significance was not confirmed in multivariate analysis (*p* = 0.8716). Notably, we observed that inflammatory markers and IL-6 levels varied heterogeneously after tocilizumab therapy, and patients with increasing levels after treatment did not present clinical or respiratory worsening. Results from the TOCIVID-19 trial, a multicenter, single-arm, open-label, phase 2 study on the efficacy and tolerability of tocilizumab in the treatment of patients with COVID-19 pneumonia (NCT04317092), suggested that tocilizumab may reduce lethality at 30 days, although its impact at 14 days seems less relevant [57]. Furthermore, an open-label, randomized, multicenter study on the efficacy of early, compared to late, administration of tocilizumab in reducing the number of patients with COVID-19 pneumonia who require mechanical ventilation is now ongoing (NCT04346355).

Abnormalities in coagulation testing have been observed in nearly 20% of patients with COVID-19 disease, and heparin has been recommended by several expert societies because of the risk of venous thromboembolism. Recently, Giardini et al. identified a possible mechanism between SARS-CoV-2 infection and the prothrombotic profile observed in these patients. Data offered a link between ACE2 downregulation and an AngII/s-Flt-1-mediated endothelial dysfunction in a model that strictly resembles preeclampsia, which could offer an explanation to the pathogenesis of the acute global vascular damage observed in these patients [58]. Several studies have suggested a high rate of thromboembolic complications among hospitalized patients, particularly those presenting with severe disease [5,59,60,61]. In a retrospective series of more than 2500 hospitalized patients with COVID-19, anticoagulation was associated with improved in-hospital survival in intubated patients, with no evident increment of bleeding events [62]. In a retrospective study of individuals with severe COVID-19, LMWH appeared to be associated with improved survival when compared with no pharmacologic prophylaxis, especially in patients with high D-dimer levels [63]. Reports from ICU patients with severe COVID-19 suggest a higher incidence of venous thrombosis embolism, even when using standard prophylaxis [64,65]. The question of an adequate anticoagulation dose for thromboprophylaxis has also been raised in COVID-19 patients. There are no actual data prospectively comparing the different levels of anticoagulation (prophylactic, intermediate, or therapeutic dosing) in COVID-19 patients, and clinical trials are ongoing. In our study population, we favor intermediate-dose anticoagulation (LMWH 100 U/Kg/d) as pharmacologic prophylaxis of venous thromboembolism for hospitalized patients with COVID-19. Interestingly, we found no difference in terms of 28-day survival between patients treated with intermediate-dose or therapeutic-dose anticoagulation for previous medical issues. In contrast, thromboprophylaxis at any dose appears to be associated with improved survival when compared with no pharmacologic prophylaxis, and significance is maintained even when combined in multivariate analysis.

## 5. Conclusions

Our study has several limitations due to the retrospective nature of the study. Overall, we found that age at the onset of SARS-CoV-2 disease is a powerful predictor of in-hospital mortality. In addition, intermediate- or therapeutic-dose anticoagulation, early short-term antiviral therapy, and short courses of corticosteroids at the end of the viral replication phase proved to be effective treatments for COVID-19 disease. Interestingly, hydroxychloroquine therapy has not been confirmed as significant in the treatment of SARS-CoV-2 infection, in accordance with the latest literature data. Finally, impairment in inflammation markers can find application as predictors of poor outcomes for hospitalized patients when incorporated into an easy-to-use prognostic score, helping clinicians to identify patients with poor prognoses at an early stage. Confirmation from a validation cohort is expected. Results from this real-life single-center experience are in agreement and confirm actual literature data on SARS-CoV-2 pneumonia in terms of both risk factors for in-hospital mortality and the effectiveness of the different therapies proposed for the management of COVID-19 disease. Results from randomized clinical trials are expected.

## Figures and Tables

**Figure 1 vaccines-09-00640-f001:**
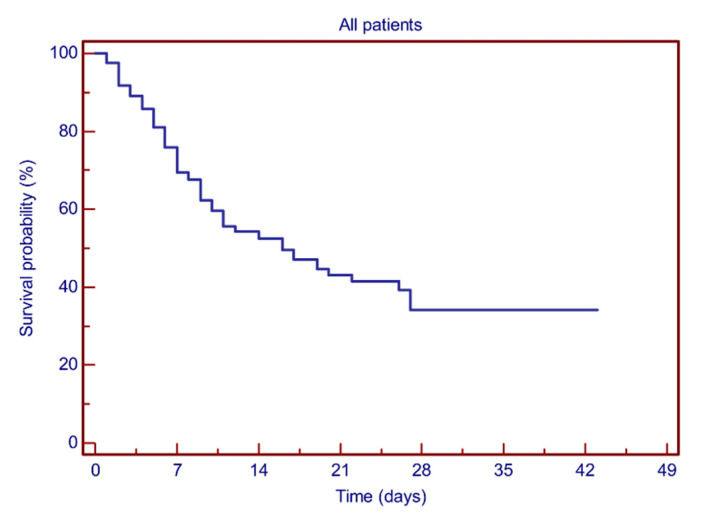
Kaplan–Meier plots showing the probability of survival for the whole patient population.

**Figure 2 vaccines-09-00640-f002:**
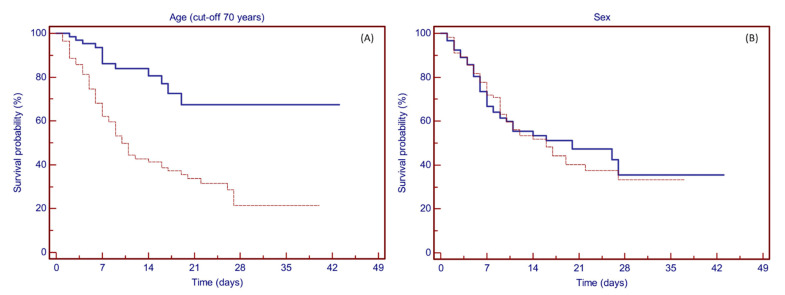
Kaplan–Meier plots showing the probability of survival based on patients’ characteristics. (**A**) Patients aged less than 70 years (solid line) or older than or equal to 70 years (dashed line). Age showed prognostic significance for survival at a cut-off point set at 70 years (*p* < 0.0001), with an estimated 28-day survival rate of 67.4% and 21.4%, respectively. (**B**) Survival was not affected by gender (female, solid line; male, dashed line) (*p* = 0.086).

**Figure 3 vaccines-09-00640-f003:**
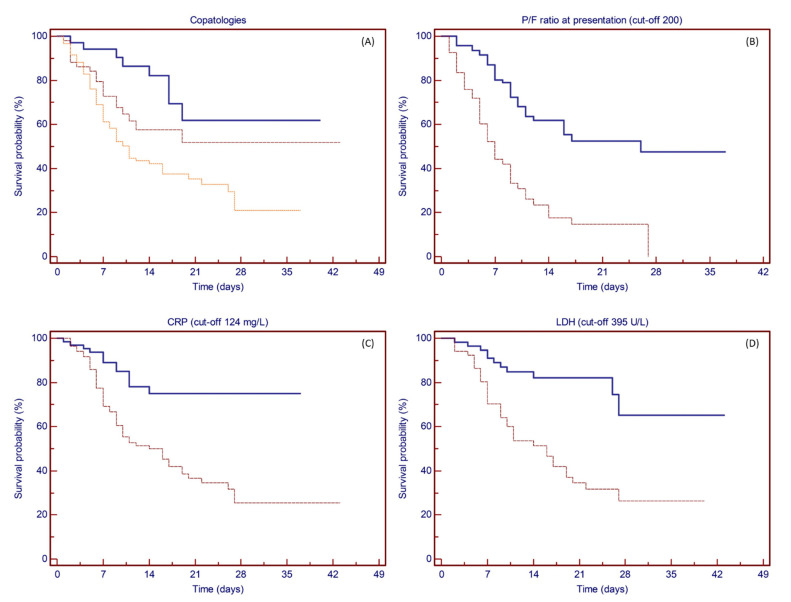
Kaplan–Meier plots showing the probability of survival based on clinical variables. (**A**) Patients with no comorbidity (solid line), 1 comorbidity (dashed line), and 2 or more comorbidities (dotted line). The number of comorbidities showed prognostic significance for survival (*p* = 0.0008), with an estimated 28-day survival rate of 61.8%, 51.7%, and 35.3%, respectively. (**B**) P/F ratio at presentation more than 200 (solid line) or less than or equal to 200 (dashed line). P/F ratio at presentation showed prognostic significance for survival at a cut-off point set at 200 (*p* < 0.0001), with an estimated 21-day survival rate of 52.4% and 14.7%, respectively. (**C**) CRP less than 124 (solid line) or more than or equal to 124 (dashed line). CRP values showed prognostic significance for survival at a cut-off point set at 124 mg/L (*p* = 0.0001), with an estimated 28-day survival rate of 74.9% and 25.4%, respectively. (**D**) LDH less than 395 (solid line) or more than or equal to 395 (dashed line). LDH values showed prognostic significance for survival at a cut-off point set at 395 U/L (*p* = 0.0001), with an estimated 28-day survival rate of 65.3% and 26.4%, respectively.

**Figure 4 vaccines-09-00640-f004:**
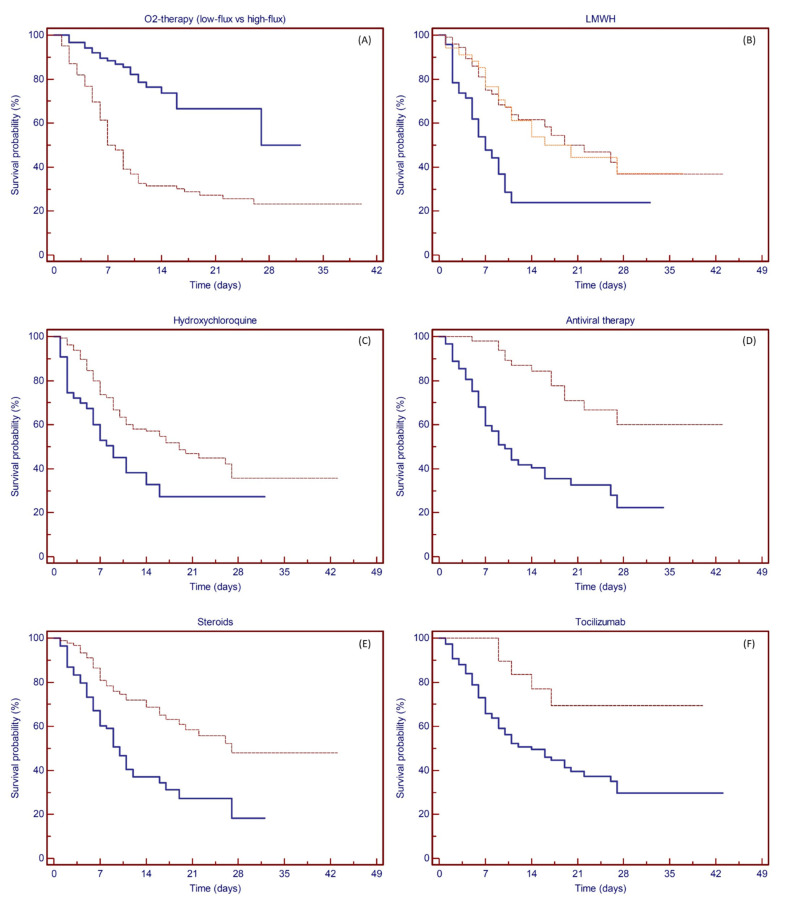
Kaplan–Meier plots showing the probability of survival based on treatments. (**A**) Patients treated with low-flux (solid line) or high-flux (dashed line) oxygen therapy. O2 therapy showed prognostic significance for survival (*p* < 0.0001), with an estimated 28-day survival rate of 50.0% and 23.3%, respectively. (**B**) No anticoagulation therapy (solid line) vs. LMWH at prophylaxis dose (dashed line) or therapeutic dose (dotted line). LMWH treatment showed prognostic significance for survival (*p* = 0.0001), with an estimated 28-day survival rate of 23.8%, 36.9%, and 37.1%, respectively. (**C**) No HCQ (solid line) vs. HCQ treatment (dashed line). HCQ showed prognostic significance for survival (*p* = 0.0029), with an estimated 28-day survival rate of 27.3% and 35.7%, respectively. (**D**) No antiviral (solid line) vs. antiviral treatment (dashed line). Antiviral therapy showed prognostic significance for survival (*p* < 0.0001), with an estimated 28-day survival rate of 22.4% and 60.1%, respectively. (**E**) No steroid (solid line) vs. steroid treatment (dashed line). Steroid therapy showed prognostic significance for survival (*p* < 0.0001), with an estimated 28-day survival rate of 18.2% and 47.9%, respectively. (**F**) No tocilizumab (solid line) vs. tocilizumab treatment (dashed line). Tocilizumab therapy showed prognostic significance for survival (*p* = 0.0059), with an estimated 28-day survival rate of 29.6% and 69.4%, respectively.

**Figure 5 vaccines-09-00640-f005:**
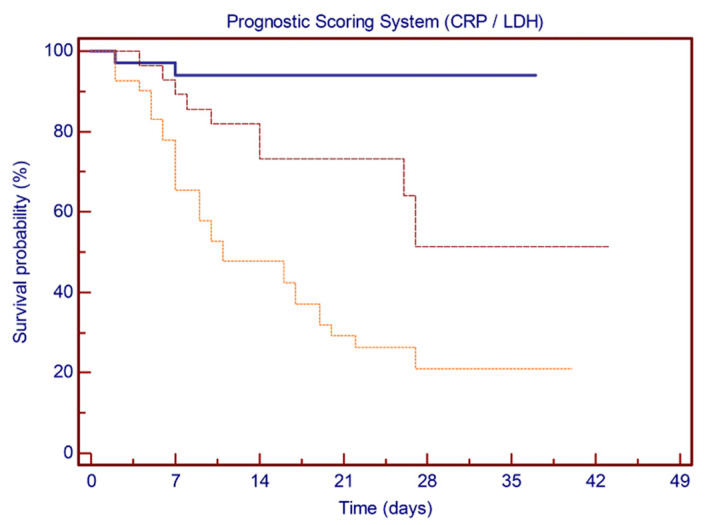
Kaplan–Meier plots showing the probability of survival based on the prognostic score model. Patients presenting none (solid line), one (dashed line), or both (dotted line) of the positive markers. Estimated 28-day survival rate varies from 94.0% to 51.3% and to 21.0%, respectively (*p* < 0.0001).

**Table 1 vaccines-09-00640-t001:** Clinical characteristics at presentation of patients with SARS-CoV-2 disease.

Variable	All Patients	Male	Female
Patients, no	205	113	92
Median age, y (IQR)	77 (18)	76 (20)	78 (15)
Median P/F ratio (IQR)	242 (124)	240 (115)	248 (138)
Median WBC, x 109/L (IQR)	7.2 (4.9)	7.2 (4.8)	6.7 (4.6)
Median LDH, U/L (IQR)	339 (170)	364 (198)	294 (139)
Median CRP, mg/L (IQR)	98 (121)	100 (120)	82 (109)
Median PCT, µg/L (IQR)	0.15 (0.29)	0.16 (0.30)	0.14 (0.23)
Median ALT, U/L (IQR)	27 (28)	33 (32)	21 (20)
Median AST, U/L (IQR)	39 (29)	43 (36)	36 (24)

ALT: alanine aminotransferase (normal value: <50); AST: aspartate aminotransferase (normal value: <50); CRP: C-reactive protein (normal value: <5); IQR: interquartile range; LDH: lactate dehydrogenase (normal value: <248); PCT: procalcitonin (normal value: <0.10); P/F ratio: ratio of arterial oxygen partial pressure to fractional inspired oxygen; WBC: white blood cells (normal value: 4.0–10.8).

**Table 2 vaccines-09-00640-t002:** Multivariate analysis.

Variable	b	SE	Exp (b)	95% CI of Exp (b)	*p*
Age (cut-off 70y)	1.1821	0.3011	3.2612	1.8129 to 5.8665	<0.0001
Copatologies	0.2524	0.1830	1.2871	0.9009 to 1.8390	0.1678
Anticoagulation	−0.3736	0.1692	0.6883	0.4948 to 0.9574	0.0273
Antiviral therapy	−1.2202	0.3106	0.2952	0.1611 to 0.5409	<0.0001
Steroid therapy	−0.5919	0.2347	0.5533	0.3501 to 0.8744	0.0117
Tocilizumab therapy	0.0858	0.5311	1.0896	0.3868 to 3.0692	0.8716
Hydroxychloroquine therapy	−0.3409	0.2345	0.7111	0.4502 to 1.1234	0.1460

## Data Availability

The data presented in this study are available on request from the corresponding author.

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
