# Peer review of "Clinical Course and Risk Factors for In-Hospital Mortality of 205 Patients with SARS-CoV-2 Pneumonia in Como, Lombardy Region, Italy"

_vaccines, 2021, doi:10.3390/vaccines9060640_

Round 1

Reviewer 1 Report

Dear Authors
I appreciate the presentation of this manuscript COVID19 has a significant impact on life, and scientific knowledge is still scarce.
Real-life data is extremely important, particularly because there are still no RCTs that provide scientific evidence I think the work should be published I just found a letter exchange on the line 319 (COVID)

Keep safe
Many regards

Author Response

Thank you for the opportunity to revise my paper. I have corrected the errors noted on the line 319.

Reviewer 2 Report

This paper investigates the risk factors for in-hospital mortality and the effectiveness of different treatment strategies on 205 laboratory-confirmed cases of COVID-19 during the Lombardy outbreak in Italy. The work is interesting, clear, and well-written. The introduction provides enough background. the objective is clear enough. The Methods section provides sufficient details. The Results and Discussion are correctly described. I only have a minor suggestion of adding a section for conclusions to summarize the major findings of the study.

There was no section for conclusions at the end of the discussion or separately to summarize the  major findings of the paper. I suggest adding a section for conclusions.

Author Response

Thank you for the opportunity to revise my paper. As a response to your comment, the Conclusions section has been added at the end of the discussion to summarize the major findings of the paper.

Reviewer 3 Report

The paper is interesting and well written. However, I suggest to improve the article to briefly discuss the incidence of SARS-Cov2 infections in patients with autoimmune diseases (see and add as reference paper by Ferri et al concerning a clinical study on italian rheumatic patients)

Author Response

Thank you for the opportunity to revise my paper. I agree with your comment concerning the opportunity to implement the introduction with a note on incidence and severity of SARS-Cov2 infections in immunocompromised patients. I have implemented references with the paper you suggested.

Reviewer 4 Report

Ref: vaccines-1226380

Title: No Clinical course and risk factors for in-hospital mortality of 205 2 patients with SARS-CoV-2 pneumonia in Como, Lombardy Region, Italy.

Journal: Vaccines

Reviewer’s comments

The paper is really interesting, addressing an actual topic using a robust statistical approach. The study is clearly described and the results found are noteworthy. So, in my opinion the paper is suitable for publication. I report only some minor assessments that are necessary to the manuscript before its publication.

Specific points:

Introduction

  • Lines 49-50: please refer to a specific date in stating “a viral disease infecting more than six million individuals all over the world”, as at the moment the number is dramatically higher; or you can update the data reported.
  • Lines 59-60: “median incubation period” seems not the correct term; maybe 2 and 12 days are the lower and upper extreme of the incubation period, respectively?
  • Lines 107-108: could you supply a reference for Tocilizumab evaluated in randomized trials?
  • Lines 108-109: similarly, is there a reference also for glucocorticoids used in patients with moderate to severe ARDS?
  • Lines 113-114: the study design should be defined more specifically than “real-life single-center cohort”, referring to the classical epidemiological studies and explaining the peculiarities of yours; moreover, here you declare two main aims of the study, but in fact you also explore the predictivity of risk factors on prognosis, supplying very interesting results in section 3.2. Prognostic score model. Maybe you can introduce and enhance also this aspect

Materials and Methods

  • In this section, a lot of statistical tests were cited, but for some of them no results are presented in the following section; please limit the description of statistical methods to what you effectively performed in this study.
  • Lines 116-118: please, if possible, specify the period of admission to hospital of patients.
  • Lines 140-155: whether or not to be assigned to the group with a specific therapy seems to be decided on the basis of clinical criteria and/or availability of the drug; this could represent a selection bias and may raise some doubts about the statistical inference on a "sample" that is not representative of any population.
  • Lines 164-165: is it sure that it is always IOT and not also INT (nasotracheal), which is usually preferred in intensive care units for long-term intubations? Maybe it is better to talk more generally about tracheal intubation and/or invasive mechanical ventilation.
  • Line 185: “p <.05” should be “p <0.05
  • Line 186: please provide a reference for MedCal statistical software, supposing its real name is MedCalc.

Results

  • Line 190: it’s better to use “min” and “max”, or “from” and “to”, instead of “range”.
  • Lines 198-205 seem to be more appropriate in Materials and Methods section, as they describe the analyses performed.
  • The same for the sentence at lines 206-207.

Discussion

  • Some parts of the Discussion could be moved to the Introduction, or at least the topic address there and then resumed in Discussion.
  • Line 357: “(p 0.146)” should be “(p=0.146)”.
  • Line 388: “risk factors” should be “risk factor”.
  • Line 423: “(p 0.8716)” should be “(p=0.8716)

Tables

  • Table 1: IQR is usually a single number, calculated as the difference between third and first quartiles.
  • Table 2: the comma separators should be converted in points for uniformity.

Author Response

Thank you for the opportunity to revise my paper. I have commented below on each of the points.

Introduction

Lines 49-50: please refer to a specific date in stating “a viral disease infecting more than six million individuals all over the world”, as at the moment the number is dramatically higher; or you can update the data reported.

I corrected with a more general sentence

Lines 59-60: “median incubation period” seems not the correct term; maybe 2 and 12 days are the lower and upper extreme of the incubation period, respectively?

I corrected the sentence

Lines 107-108: could you supply a reference for Tocilizumab evaluated in randomized trials?

I have implemented references with the paper by Salvarani et al, JAMA Intern Med. 2021, Hermine et al, JAMA Intern Med. 2021, and Stone et al, N Eng J Med. 2020

Lines 108-109: similarly, is there a reference also for glucocorticoids used in patients with moderate to severe ARDS?

I have implemented references with the meta-analysis by Siemieniuk R et al, BMJ 2020

Lines 113-114: the study design should be defined more specifically than “real-life single-center cohort”, referring to the classical epidemiological studies and explaining the peculiarities of yours; moreover, here you declare two main aims of the study, but in fact you also explore the predictivity of risk factors on prognosis, supplying very interesting results in section 3.2. Prognostic score model. Maybe you can introduce and enhance also this aspect

I corrected the sentences as suggested

Materials and Methods

In this section, a lot of statistical tests were cited, but for some of them no results are presented in the following section; please limit the description of statistical methods to what you effectively performed in this study.

The tests indicated in the paragraph between lines 172 and 180 are the descriptive and basic methods of analysis of the data in question. They represent the basic statistical analysis and validation of data on which we later carried out the remaining univariate and multivariate statistical analysis. If you confirm there is no need to specify these basic analysis I can remove the selected paragraph.

Lines 116-118: please, if possible, specify the period of admission to hospital of patients.

I corrected the sentences as suggested

Lines 140-155: whether or not to be assigned to the group with a specific therapy seems to be decided on the basis of clinical criteria and/or availability of the drug; this could represent a selection bias and may raise some doubts about the statistical inference on a "sample" that is not representative of any population.

I specify that patients received specific interventions on the base of their clinical needs and comorbidities.

Lines 164-165: is it sure that it is always IOT and not also INT (nasotracheal), which is usually preferred in intensive care units for long-term intubations? Maybe it is better to talk more generally about tracheal intubation and/or invasive mechanical ventilation.

In our Centre patients were treated with IOT when admitted to ICU. Anyway, I change the sentence in” invasive mechanical ventilation”.

Line 185: “p <.05” should be “p <0.05”

I corrected the sentences as suggested

Line 186: please provide a reference for MedCal statistical software, supposing its real name is MedCalc.

I confirm that MedCal is the statistical software we used in analysis. I modified the citation as indicated at the productor site: https://www.medcalc.org/faq/027.php

Results

Line 190: it’s better to use “min” and “max”, or “from” and “to”, instead of “range”.

We used the term "range" in the statistical sense, precisely to indicate the lowest and highest value of all observations.

Lines 198-205 seem to be more appropriate in Materials and Methods section, as they describe the analyses performed.

The same for the sentence at lines 206-207.

I moved the paragraphs indicated in the Materials and Methods section

Discussion

Some parts of the Discussion could be moved to the Introduction, or at least the topic address there and then resumed in Discussion.

Line 357: “(p 0.146)” should be “(p=0.146)”.

Line 388: “risk factors” should be “risk factor”.

Line 423: “(p 0.8716)” should be “(p=0.8716)”

I corrected the sentences as indicated.

Tables

Table 1: IQR is usually a single number, calculated as the difference between third and first quartiles.

Table 2: the comma separators should be converted in points for uniformity.

I corrected the sentences as indicated.